# Effect of Peony (*Paeonia ostii*) Seed Meal Supplement on Enzyme Activities and Flavor Compounds of Chinese Traditional Soybean Paste during Fermentation

**DOI:** 10.3390/foods12173184

**Published:** 2023-08-24

**Authors:** Weiqi Fu, Jiamin Ren, Shuwen Li, Dirong Ren, Xixi Li, Chenghuan Ren, Xueru Zhao, Jiaying Li, Fengjuan Li

**Affiliations:** State Key Laboratory of Food Nutrition and Safety, Tianjin University of Science & Technology, Tianjin 300457, China; fwqtust@mail.tust.edu.cn (W.F.); renjm2023@163.com (J.R.); 22845940@mail.tust.edu.cn (S.L.); renserein@mail.tust.edu.cn (D.R.); s15145220@163.com (X.L.); 22845937@mail.tust.edu.cn (C.R.); 15227439583@163.com (X.Z.); lijiaying724@163.com (J.L.)

**Keywords:** peony seed meal, fermented soybean paste, enzyme activity, koji making, flavor compounds

## Abstract

Peony seed meal (PSM) is the by-product obtained from peony seeds after oil extraction. In this study, PSM was incorporated into traditional koji-making, and its impacts on koji enzyme activities and flavor compounds in final products were investigated. In the process of koji fermentation, the optimal addition ratio of PSM to soybean was determined as 7:3. Under this ratio, the maximum enzyme activities of neutral protease, amylase, and glucoamylase were 1177.85, 686.58, and 1564.36 U/g, respectively, and the koji obtained was subjected to maturation. During post-fermentation, changes in the fermentation characteristics of the paste samples were monitored, and it was found that compared to the soybean paste without PSM, the enzyme activities maintained at a relatively good level. The PSM soybean paste contained a total of 80 flavor compounds and 11 key flavor compounds (OAV ≥ 1), including ethyl isovalerate, isovaleric acid, hexanal, phenylacetaldehyde, 3-Methyl-1-butanol 4-heptanone, 2-pentylfuran, methanethiol ester caproate, isoamyl acetate, 3-methyl-4-heptanone, and isovaleraldehyde. These findings could be used to improve the quality of traditional fermented paste, enrich its flavor, and simultaneously promote PSM as a valuable resource for fermented foods.

## 1. Introduction

The utilization of agricultural by-products, particularly the cake or meal obtained after oil extraction from oil-bearing crops, has garnered significant attention from researchers worldwide due to its immense potential for food processing and sustainable development [1]. For instance, the meal of pumpkin [2] or flaxseed [3] has already been applied in the production of food or food additives through the process of solid-state fermentation. Peony (*Paeonia ostii*), a deciduous shrub belonging to the genus Paeonia in the family Paeoniaceae, is known for its seeds that are abundant in oil, protein, flavonoids, paeoniflorin, and other active ingredients. Given the notable functional properties of peony seed oil, the cultivation of oil peonies and related industries in China have experienced significant growth in recent years, resulting in an annual increase in peony seed meal (PSM) production. PSM is a by-product derived from the oil extraction process of peony seeds, which contain substantial amounts of proteins, polysaccharides, vitamins, minerals, and other substances [4,5]. PSM extract exhibits various wholesome effects, such as antioxidant and anti-cancer properties [6,7]. Traditionally, PSM has been utilized as an alternative nitrogen source in livestock feed or discarded outright; however, there was little information on its utilization in food production. The use of PSM remains relatively limited in the current market, leading to the underutilization of the abundant nutrients present in the seed meal and consequent squandering of its substantial economic value.

Soybean paste, known as miso in Japan [8], doenjang in Korea [9], and Dajiang in China [10], is a fundamental component of traditional fermented foods and holds significant importance in the cuisines of Southeast and East Asia. It is typically prepared by fermenting soybean and wheat flour together. During the fermentation process, macromolecular substances such as proteins and polysaccharides undergo hydrolysis, resulting in the conversion of these complex molecules into smaller, more easily absorbable components. This transformation enhances the nutritional profile of the paste and imparts various beneficial properties that contribute to consumer health. To enhance the nutritious quality of soybean paste and enrich its flavor, the supplementation of other food ingredients, including Tartary buckwheat [11], red ginseng marc [12] and so on, have been explored. Therefore, the fermentation of PSM-based soybean paste holds the potential for diversifying the range of fermented condiments available in the market, broadening dietary choices for consumers, and meeting their preferences for a health-conscious diet.

In this sense, the present study aims to investigate the impact of PSM addition on soybean paste’s physicochemical properties and fermentation characteristics. Specifically, the enzyme activity levels during the fermentation process and flavor compound profiles of the end products were assessed. It would be helpful to enhance the utilization value of PSM, thereby offering valuable insights and rationale for the development of other seed meal resources.

## 2. Materials and Methods

### 2.1. Materials and Chemicals

PSM, made by cold-pressed method, was purchased from Heze Guyu Peony Co., Ltd. (Heze, China). The protein, fat, ash, and moisture content was 23.25, 14.59, 9.39, and 6.18 g/100 g, respectively. Soybeans and wheat flour were purchased from Shandong Tianxialiangcang Food Co., Ltd. (Zibo, China) and Hebei Jinshahe Group (Hebei, China). *Aspergillus oryzae* 3.042 was provided by Xiangyuan Biotechnology Co., Ltd. (Jining, China). 2-Methyl-3-heptanone (≥99%) was supplied by Aladdin Co., Ltd., (Shanghai, China).

### 2.2. Koji Fermentation

A hundred grams of PSM was smashed by a pulverizer and desiccated in a drying oven at 60 °C for 15 min (Figure 1). Dehulled soybean was cleaned and soaked in water for 12 h (*w*:*w* = 1:3). Subsequently, all ingredients were steamed at 121 °C and 100 kPa for 30 min. Once the ingredients cooled to approximately 28–30 °C, PSM, soybeans, and flour were uniformly mixed in various ratios (0:10:3, 3:7:3, 5:5:3, 7:3:3, and 10:0:3, *w*/*w*). Each 100 g of mixture was inoculated with *A. oryzae* spores suspension (1 × 10^7^ spores/mL) and incubated at 30 °C for 3 d. Samples were collected at 0, 24, 48, and 72 h and stored at −20 °C for subsequent analysis.

### 2.3. Post-Fermentation

A portion of harvested koji (60 g) was transferred to a 150 mL glass jar and mixed with an equal weight of 20% (*w*/*v*) brine, resulting in the creation of the PSM soybean paste group (PSP). At the same time, sample groups of pure soybeans were prepared (SP). The entire post-fermentation process was incubated at 40 °C for 35 d. Samples were collected at 0, 1, 3, 5, 7, 14, 21, 28, and 35 d and stored at −20 °C for subsequent analysis.

### 2.4. pH Value Assay

The mixture consisted of 3 g of the ground sample and 20 mL of distilled water that was vigorously vortexed, and the pH value was measured at room temperature.

### 2.5. Neutral Protease Activity Assay

Neutral protease (EC 3.4.24.4) activity was determined by the method described by Gao et al. [13]. Briefly, 5 g of thoroughly crushed koji samples were diluted with 50 mL of distilled water and filtered through four layers of cotton gauze to extract crude enzyme. One milliliter of the filtrate was added to 1 mL of 2% (*w*/*w*) casein (2 g of casein dissolved in 100 mL of 0.1 M sodium phosphate buffer (pH 7.2)), both pre-incubating at 40 °C for 5 min, and then the mixture was incubated at 40 °C for 10 min to liberate tyrosine. After the incubation, 2 mL of 0.4 M trichloroacetic acid was added, and the mixture was incubated at 40 °C for 20 min to precipitate the residual protein. The protein was removed by centrifugation at 3500 rpm for 5 min. Then, 1 mL of the supernatant was combined with 5 mL of 0.4 M sodium carbonate and 1 mL of 1 N Folin-Ciocalteu phenol reagent (Sigma-Aldrich, St. Louis, MO, USA). After adequate vortexing, the mixture was incubated at 40 °C for 20 min. The absorbance of the mixture was recorded at 680 nm and compared to a standard tyrosine curve for quantitative analysis.
(1)$$Neutral\;protease\;activity\;(U/g)=\frac{A\times N\times 4}{10\times W}$$
where *A* is the amount of tyrosine calculated from the standard curve, *N* is the sample dilution multiple, and *W* is the sample mass in grams.

One unit (U) of neutral protease activity was defined as the amount of 1 μg tyrosine generated by the enzymatic process per gram sample per minute under the assay conditions (pH 7.2, 40 °C).

### 2.6. Amylase Activity Assay

The amylase (EC 3.2.1.1) activity assay was performed according to the 3,5-Dinitrosalicylic acid (DNS) method described by Su et al. [14].

The reaction mixture comprised 1 mL of suitably diluted enzyme solution in distilled water and 1 mL of a 1% (*w*/*v*) soluble starch solution. Both components were pre-incubated at 40 °C for 5 min. The mixture was heated for 5 min in a boiling water bath and then cooled to room temperature. The resulting solution’s absorbance was measured at 540 nm using a UV spectrophotometer (UV-1800, Mapada Ltd., Shanghai, China), and the maltose concentration was calculated based on the maltose calibration curve. One unit (U) of amylase activity was defined as the amount of maltose hydrolyzed in micromoles per gram of sample within 1 h under the aforementioned assay conditions.

### 2.7. Glucoamylase Activity Assay

The method for determining glucoamylase (EC.3.2.1.3) activity was described by Fadel et al. [15] with appropriate modifications.

The crude koji enzyme solution was diluted 2.5 folds with distilled water. Next, 500 μL of a 1% (*w*/*v*) soluble starch solution was mixed with 400 μL of acetic acid-sodium acetate buffer (pH 4.6) and preheated at 40 °C for 5 min. Subsequently, 200 μL of the diluted enzymes were added to the mixture and incubated at 40 °C for 10 min. After that, 20 μL of 1 M NaOH was added to terminate the reaction. A mixture of 60 μL and 200 μL of DNS reagent was incubated in boiling water for 5 min to develop color, followed by immediate cooling to room temperature. A volume of 20 μL of the diluted reaction mixture was extracted and analyzed using a microplate reader (Infinite M200 PRO, Tecan Group Ltd., Männedorf, Switzerland) at 540 nm. For the control groups, the enzymes were inactivated first by adding 20 μL of 1 M NaOH. The glucose concentration was determined using the glucose calibration curve. One unit (U) of glucoamylase activity was defined as the amount of glucose generated per hour under the specified conditions (pH 4.6, 40 °C).

### 2.8. Determination of Reducing Sugar

The reducing sugar content was measured using the DNS method according to Kaewkrod et al. [16].

Briefly, 60 μL of aqueous extract was mixed with 200 μL of DNS solution and heated in boiling water for 5 min. After appropriate dilution, the mixture was measured at 540 nm using a spectrophotometer. The reducing sugar content was calculated by a glucose calibration curve measured and expressed as g/100 g dry weight.

### 2.9. GC-MS Analysis of Volatile Compounds

The volatile flavor compounds of the soybean paste sample were analyzed using headspace solid-phase microextraction (HS-SPME) combined with gas chromatography-mass spectrometry (GC-MS), following the method adapted from Yue Yang et al. [17]. Each ground paste sample (5 g) was mixed with 5 mL of milli-Q water in a 20 mL SPME vial, and 5 μL of 2-methyl-3-heptanone (0.16 μg/kg, dissolved in methanol) was applied as the internal standard. The flavor compounds were extracted from the mixture using a divinylbenzene/carboxen/polydimethylsiloxane (DVB/CAR/PDMS) SPME fiber (50/30 μm, Supelco Inc., Bellefonte, PA, USA) under stirring conditions (800 rpm) at 70 °C for 30 min, following by a 10 min pre-incubation. The volatile compounds were qualitatively analyzed using a QP-2010 GC-MS instrument equipped with a TG-WAXMS capillary column (30 m × 0.25 mm, 0.25 μm, Agilent Technologies, Santa Clara, CA, USA). The extraction fiber was inserted into the injection port and desorbed for 15 min. Helium gas was used as the carrier gas at a constant flow rate of 0.75 mL/min with a split ratio of 20:1. Programmed ramp-up conditions were as follows: oven temperature started at 40 °C and was maintained for 3 min, ramped up to 180 °C at 5 °C/min and was maintained for 1 min, then raised to 250 °C at 7 °C/min and was held for 5 min. The mass detector operated in electronic impact (EI) mode at 70 eV, with a scanning range of 40–500 *m*/*z*. Both the ion source and interface were maintained at 220 °C.

Qualitative analysis was conducted by comparing the results with the National Institute of Standards and Technology database file (NIST11/11s) and using the retention index (RI). A compound was identified if it had at least 80% similarity to the database and a RI value close to the published literature.

The semi-quantitative analysis of each compound was performed by the internal standard method.

### 2.10. Calculation of Odor Activity Values (OAVs)

The odor activity values (OAVs) are an index used to assess odor potency and are the ratio of the concentration of volatile compounds to the detection threshold. In general, only flavor substances with OAV ≥ 1 contribute to the overall flavor profile of soybean paste while emitting aroma, and these compounds are referred to as aroma active compounds. The OAV is calculated as follows:OAV = C/T(2)
where C is the concentration of volatile compounds in μg/kg, and T is the aroma threshold value, obtained by reviewing the literature in μg/kg.

### 2.11. Statistical Analysis

All the experiments were set up in three parallels (n = 3), and the results were expressed as mean ± standard deviation. Statistical analyses were performed using IBM SPSS Statistics software (version 26, IBM Corp., Armonk, NY, USA). Significant differences between samples were assessed using a one-way analysis of variance (ANOVA) and Duncan’s test. *p* < 0.05 was considered as a significant difference between samples, and data were plotted using Origin Pro 2023 software (Northampton, MA, USA.).

## 3. Results and Discussion

### 3.1. Fermentation Characteristic Changes in Paste Samples during Koji Fermentation

The activity of the protease produced by *A. oryzae* was influenced by the content and type of nitrogen source present in the environment. As shown in Figure 2a, the neutral protease enzyme activity in koji, produced from various ratios of raw materials, gradually increased in the first 48 h, followed by a decline after reaching its peak at 48 h. The maximum enzyme activity of 1177.85 U/g was observed in the sample with the 7:3 soybean-to-PSM ratio.

Amylase plays a key role in the koji fermentation process, primarily hydrolyzing the starch present in the raw material into smaller molecules, including dextrin, maltose, and trace amounts of glucose. The enzyme activity level determines the final product’s reducing sugar content and mouthfeel. According to Zhao et al. [18], the study revealed that protease and amylase work synergistically, leading to the formation of diverse flavors, including alcohols, acids, esters, aldehydes, furans, and pyrazines, during the fermentation process. The effects of different raw material ratios on amylase enzyme activity during the koji fermentation process are shown in Figure 2b. The amylase activity in the samples exhibited an increasing trend as fermentation progressed, and the sample with a soybean-to-PSM ratio of 7:3 reached its peak enzyme activity at 72 h, measuring 686.58 U/g.

In the koji fermentation stage, glucoamylase hydrolyzes starches into glucose and oligosaccharides. These substances contribute to microbial growth and the Maillard reaction, ultimately shaping the physicochemical characteristics of soybean paste. Figure 2c depicts the progressive increase in glucoamylase activity during the koji fermentation process. Additionally, all samples supplemented with seed meal achieved peak glucoamylase enzyme activity at 72 h, with values of 1564.36, 859.05, 906.48, and 365.85 U/g, respectively (Figure 2c). The sample with a soybean-to-PSM ratio of 7:3 exhibited the highest glucoamylase activity.

Figure 2d demonstrates a significant increase in reducing sugar content within the first 24 h of fermentation in samples utilizing various raw material ratios. This increase positively correlated with the activities of amylase and glucoamylase during fermentation. Subsequently, after 24 h, the reducing sugar content progressively declined, which was attributed to the stabilization of enzyme activity and its utilization by microbial growth. Compared to other samples, the faster decline in reducing sugar content in samples with soybean-to-PSM ratios of 10:0 and 7:3 could be attributed to the heightened metabolic activity of microorganisms, leading to increased consumption of reducing sugars.

### 3.2. Fermentation Characteristic Changes in Paste Samples during Post-Fermentation

The pH of the two samples exhibited a consistent decreasing trend (Figure 3a) that could be attributed to the accumulation of organic acids during the post-fermentation process. During the initial phase of post-fermentation, *Aspergillus* in the koji converts sugars, proteins, and other raw material components into organic acids, acidic amino acids, and more. Adequate quantities of these substances not only enhance the flavor of the paste but also contribute to the extended shelf life of bean paste. Furthermore, as fermentation progresses, these organic acids can undergo esterification with the alcohols produced during fermentation, and this reaction leads to a significant stabilization of the pH and prevents it from further decreasing [19].

During the fermentation process, the activities of all three enzymes decreased to varying extents, with the neutral protease exhibiting the most significant decrease in activity. The pH changes observed during the post-fermentation process (Figure 3a) indicated that the accumulation of acidic substances in the paste resulted in a progressive decrease in the environmental pH during fermentation. This decrease in pH had a more significant impact on the activity of the neutral protease, which has an optimum pH range of 6.0–7.5. Moreover, the protease secreted by *A. oryzae* is an inducible enzyme, meaning that its secretion diminishes when the protein content in the environment decreases due to hydrolysis reaction depletion. Consequently, both factors collectively contributed to a substantial decline in neutral protease activity. The PSP samples displayed lower overall neutral protease activity (Figure 3b), which experienced a more pronounced decrease during the post-fermentation process compared to the SP samples.

In both sample groups, the activities of amylase and glucoamylase exhibited a declining trend during post-fermentation, as depicted in Figure 3c,d. Compared to the neutral protease, amylase and glucoamylase, secreted by *A. oryzae*, demonstrated a lower optimum pH, rendering them less susceptible to pH changes. However, their activities gradually diminished as the substrate carbohydrates of the hydrolysis reaction were depleted in the environment. Additionally, a similar decreasing trend in both amylase and glucoamylase activities was observed within the same samples, suggesting a possbile regulatory relationship between them.

Figure 3e illustrated the changes in reducing sugar content during post-fermentation across the two sample groups. In both samples, the reducing sugar content exhibited a pattern of initial increase followed by a subsequent decrease. The maximum values were attained at 7 d (12.99 g/100 g DW) and 21 d (10.68 g/100 g DW) for each group. The PSP exhibited a higher content of 9.66 g/100 g DW between the two samples at 35 d.

### 3.3. Flavor Analysis of Paste Samples

#### 3.3.1. Volatile Flavor Substances of Paste Samples

In the production of fermented soybean products, the material composition and flavor of the raw material itself can affect the final product’s flavor to some extent. Consequently, volatile flavor substances in PSM were analyzed to assess the impact of adding PSM on the flavor of soybean paste.

As shown in Table 1, the analysis revealed the presence of 17 volatile flavor substances in the raw material of PSM, including *N*-(2-methoxyethyl)methylamine, isoamyl aldehyde (chocolate, peach), hexanal (grass, fruity), *trans*-cyclohexene carbonate (banana, cheese), 2-hexenal, *cis*-2-penten-1-ol (fruity), 2-ethyl 4-methyl 1-pentanol, (*E*,*E*)-2,4-hexadienal (floral, citrus), acetic acid, benzaldehyde (almond, cherry), methyl benzoate (holly oil, almond), 2-aminoethyl isopropyl ether, 4,7-dimethylbenzofuran (hawthorn, mimosa), benzyl alcohol (rose), benzoic acid (fat), 18-crown ether-6, and palmitic acid. Among these, the five substances with the highest content were benzoic acid (587.58 μg/kg), 2-hexenal (436.15 μg/kg), benzyl alcohol (230.70 μg/kg), 4,7-dimethylbenzofuran (183.26 μg/kg), and benzaldehyde (128.79 μg/kg). These substances primarily contributed to fruit, floral, and fat flavors. The abundance of aromatic compounds in PSMs might be attributed to the transformation of intermediate products resulting from the oxidative decomposition of plant polysaccharides [20]. These products underwent a series of conversions within the branching acid metabolic pathway or phenylalanine secondary metabolism [20,21,22], ultimately leading to the production of aromatic secondary metabolites. Also known as leaf aldehyde, 2-hexenal is the primary source of aroma in plant leaves and can be used as a safe food additive to impart strong fruit and light flavors to foods [22,23]. Aldehydes, the predominant volatile organic compounds found in vegetable oils, play a significant role in enhancing the aroma profile of peony seed oil [20]. These aldehydes are generated through three primary pathways: direct release upon disruption of plant cells, the lipoxygenase pathway (where lipoxygenase interacts with polyunsaturated fatty acids), or through autoxidation reactions [20]. 4,7-Dimethylbenzofuran was also detected in volatile flavor substances of peony seed oil, and its formation might be produced by the spontaneous oxidation of fatty acids [20].

Figure 4 revealed 115 volatile compounds in both the SP and PSP samples. Appendix A presents the results of the qualitative and semi-quantitative analysis of specific flavor substances, and Figure 5 displays the speculated formation pathways of the flavor compounds. An examination of Figure 4 reveals that the composition of flavor compound types in both the SP and PSP samples was similar, primarily comprising alcohols, acids, esters, aromatics, pyrazines, and aldehydes compounds. However, variations existed in the quantities of each compound type, with the PSP samples exhibiting more volatile compounds than the SP samples. The SP samples yielded 65 detected flavor substances, comprising 3 aldehydes, 1 ketone, 11 alcohols, 8 acids, 13 esters, 18 aromatics, 2 furans, 5 pyrazines, 1 alkane, and 3 other compounds. The PSP samples yielded a total of 80 detected flavor substances, encompassing 3 aldehydes, 4 ketones, 10 alcohols, 7 acids, 10 esters, 24 aromatics, 4 furans, 3 pyrazines, 9 alkanes, 1 olefin, and 5 other compounds. The development of the soybean paste flavor arises from the synergistic effects of multiple volatile small molecules. The primary formation pathways encompass fat oxidation, the Maillard reaction, protein degradation, amino acid conversion, starch saccharification, and alcohol fermentation.

During the initial stages of koji-fermentation and post-fermentation, the proteins present in the raw material undergo hydrolysis by microorganism-secreted enzymes, forming peptides and amino acids. The peptides function as active and taste-contributing substances, whereas the amino acids are catabolized and metabolized in the later stages of post-fermentation. This metabolic process leads to the generation of amines or the conversion of amino acids to aldehydes through the Maillard reaction. Additionally, specific amino acids can partake in Strecker degradation reactions, producing alcohols, aldehydes, and ketones. Lipoxygenase-induced enzymatic reactions are also considered crucial for the formation of flavor compounds in paste [23].

The SP samples exhibited a higher concentration of ester compounds compared to the PSP samples. Esters play a significant role in contributing to the aroma of soybean paste and are primarily generated through condensation reactions involving alcohols and acids [24].

Both samples exhibited the presence of Ethyl isovalerate (apple, pineapple), methyl palmitate (iris, oil wax), and methyl linolenic acid. However, the SP sample uniquely contained Ethyl caprate (pear, brandy), methyl caprylate (herbaceous, orange), methyl laurate, methyl linoleate (oily), methyl *trans*-9-octadecenoate, 2-methylbutyrate-3-methylbutyrate (blueberry, apple), vinyl caproate, ethyl valerate (fruity), isoamyl isovalerate (apple, banana), and ethyl 2-methylbutyrate (green apple), which contribute a fruity and sweet aroma to the product. On the other hand, Vinyl acetate, isoamyl acetate (banana), methyl nonanoate (fruity, wine), Butanoicacid *tert*-butyl ester, methyl thiol hexanoate (cocoa, coffee), and 2-methyl butyl propionate (fruity, rum) were exclusively found in the PSP sample.

The higher diversity of esters in the SP samples may be attributed to the accumulation of fatty acid glycolipids during fermentation. Fatty acid esters are most likely formed as a result of the interaction between alcohols and acids generated through metabolic processes [25].

Alcohols play a dual role in soybean paste, contributing both herbal and floral aromas and participating in the production of esters through interactions with acids. These esters enhance the aroma and taste of the final product. Although short-chain alcohols, with their higher threshold, may not impart any discernible flavor to the product, long-chain alcohols, particularly branched alcohols like 1-octanol (lemon), 2-ethylhexanol (floral), and isoamyl alcohol (whiskey, banana), were detected in both SP and PSP samples and significantly contribute to the aromatic profile of soybean paste due to their lower threshold. The SP sample exclusively contained Cyclohexanol (camphoraceous), *n*-hexanol (green, leafy), heptanol (vanilla), 3-methyl-1-hexanol, *trans*-2-octen-1-ol (citrus and vegetable scent), 1-octen-3-ol (mushroom), (*Z*)-6-nonen-1-ol (oily, melon), and 3-pentanol (herbal). Undecanol, 2-methylcyclopentanol, cyclobutanol, 1,2,3-butanetriol, 1-hexen-3-ol, 2-ethylhex-2-enol, and 3-methyl-1-penten-3-ol were unique to the PSP sample.

Each sample contained three aldehydes. Nonanal (rose and citrus) and 2,2-dimethyl-3-hydroxypropionaldehyde were exclusively detected in the SP samples. On the other hand, isovaleraldehyde (chocolate, fatty) and furfural (almond, toast) were found solely in the PSP samples. It has been documented that furfural, at pH ≤ 7, is primarily formed through the cyclized dehydration of pentose [26].

There was a notable disparity in the ketone content between the two samples. Ketones, known for their floral and fruity aromas, typically result from the microbial fermentation process involving lipid and amino acid degradation. The SP sample contained only one ketone, 1-hepten-3-one. In contrast, the PSP sample encompassed four ketones: 4-heptanone (cheesy), 3-methyl-4-heptanone (apple juice, hazelnut), 3,4-dimethyl-2-hexanone, and isopropyl tertiary butyl ketone.

The acid composition of the SP and PSP samples was similar, with the detection of isooctanoic acid, isovaleric acid (cheese), palmitic acid (slight waxy), acetic acid, and isobutyric acid. Hexanoic acid, capric acid, and 3,3-dimethylacrylic acid were exclusively found in the SP sample, whereas 3,3-dimethyl-1-butanoic acid and 2-ethylhexanoic acid (licorice, walnut) were unique to the PSP sample.

The SP samples exclusively contained 2-*tert*-butoxytetrahydrofuran, and the PSP samples contained 2-acetylfuran (cocoa, coffee), 2-ethylfuran (coffee, nutty), and 2-methylfuran (chocolate), which contributed to a caramel and roasted aroma. Furans are produced through lipid autoxidation or carbohydrate degradation, and acetylated furans have been identified in various nuts and their oils [22].

The inclusion of PSM increased the content and variety of alkane compounds in the soybean paste, resulting in the most significant difference in alkane composition between the SP and PSP samples. Alkanes typically exhibit waxy flavors; however, they have high aroma thresholds, so although multiple alkanes were identified in the PSP samples, their overall impact on aroma was minimal.

The type and amount of alkane compounds exhibited the most significant variation among volatile compounds between the two samples, and the inclusion of PSM resulted in an augmentation of alkane compounds in the soybean paste product. Although multiple alkanes with waxy flavors were detected in the PSP samples, their overall contribution to the aroma was low due to their high aroma thresholds. Alkanes are naturally occurring in plants and formed when lipids are hydrolyzed by lipases during fermentation, resulting in the production of free fatty acids. Unsaturated fatty acids undergo decarboxylation reactions through the combined action of multiple enzymes, representing an alternative pathway for the formation of ketones, aldehydes, and alcohols [27]. Conversely, PSM contains a significant amount of unsaturated fatty acids, potentially leading to variations in the specific composition and content of alkanes, ketones, aldehydes, and alcohols between the PSP and SP samples. Alkanes are considered endogenous to plants, formed when lipases hydrolyze lipids during fermentation to produce free fatty acids, of which the unsaturated fatty acids undergo decarboxylation reactions under the combined action of multiple enzymes. This is an alternative pathway for forming ketones, aldehydes, and alcohols [27]. In contrast, PSM contained a large amount of unsaturated fatty acids, which may have led to the differences in the specific composition and content of alkanes, ketones, aldehydes, and alcohols between PSP and SP samples.

Pyrazines, which have a strong aroma of roasted potatoes, fried peanuts and nuts, are particularly important for the flavor of fermented foods, especially soybean paste, and are mainly produced by the Maillard reaction, microbial metabolism, or amino acid catabolic pathway. Both samples contained 2,6-dimethylpyrazine (roasted meat, coffee), 2,3,5-trimethylpyrazine (roasted potatoes, nutty), and 2-methyl-6-vinyl pyrazine. The SP sample exhibited 2,5-dimethylpyrazine (fried peanuts, chocolate) and 2-ethyl-6-methylpyrazine (camphor, menthol).

The SP samples contained specific aromatic compounds such as methyl phenylacetate (sweet, honeyed), phenylacetic acid, homo-trimethylbenzene, 3,5-di-*tert*-butylphenol, 3-phenylfuran, and (±)-1-phenyl-2-propanol (rose).

The PSP samples exhibited the presence of various benzene derivatives, including p-xylene, which has been identified as a natural constituent of plant material [27]. Additionally, m-xylene and o-ethyltoluene are produced through the degradation of lignin or tannin [26]. PSP samples contained more aromatic compounds than SP samples due to the elevated levels of benzoic acid in the raw material of PSM. Benzoic acid, possessing properties of both aromatic and carboxylic acid compounds, can undergo substitution reactions on the benzene ring as well as esterification reactions with other organic acids and compounds.

Pyrroles were detected in both sample species and could potentially have resulted from the interaction between the reducing sugars and amino acids [28].

Only 3,5,5-trimethyl-1-hexene (hazelnut) was identified as an olefin among both samples. The limited presence of olefins might be attributed to their susceptibility to oxidative degradation in specific conditions or their involvement in forming heterocyclic compounds as intermediates [29].

#### 3.3.2. OAVs Analysis of Paste Samples

Generally, volatile compounds with an OAV ≥ 1 are present in small quantities in foods. However, their low flavor threshold allows for easy detection by the human sense of smell, classifying them as key volatile flavor compounds. Despite their low abundance, key aroma compounds should receive special attention due to their significant impact on the overall sensory acceptability of fermented food products. Table 1 presents the detection of three essential flavor substances in the raw material of PSM. Among them, methyl benzoate (OAV = 2.10) and benzaldehyde (OAV = 2.07) are considered to be the main aroma compounds of flowers, such as Eriobotrya japonica and Paeonia ludlowi, with strong almond and cherry flavors [23]. (*E*,*E*)-2,4-hexadienal (OAV = 2.33) has been identified in various plant flowers [24], fruits [25], and seafood products [26], playing a crucial role in the development of floral and fruity aromas.

Table 2 displays the detection of 18 volatile flavor substances with OAVs ≥ 1 in both SP and PSP samples, with 12 compounds found in SP samples and 11 compounds in PSP samples. The key flavor compounds shared by both samples included ethyl isovalerate, isovaleric acid, hexanal, phenylethylaldehyde, and isoamyl alcohol. Ethyl isovalerate, known for its apple and pineapple aroma, is primarily formed during the koji-fermentation stage and serves as a prominent aroma compound in traditional Chinese Zhuhoujiang and horse bean-chili-paste [30,31].

In the study of Lu et al. [30], the impact of specific volatile compounds on the overall aroma of Zhuhoujiang was investigated using aroma recombination and omission experiments. Their findings revealed that the omission of ethyl isovalerate significantly influenced the overall aroma profile of the sample. During the post-fermentation stage, the formation of flavor compounds is closely linked to a series of biochemical reactions associated with the metabolism of amino acids and proteins present in the raw material. Phenylacetaldehyde, for instance, can be generated through the metabolic processes of amino acids and undergo oxidation-reduction reactions to form corresponding acids and alcohols [32]. These compounds can then participate in esterification reactions, producing esters, which are recognized as crucial flavor compounds in soybean paste [33].

Isovaleric acid, known for its pivotal role in imparting the cheese aroma to fermented foods such as cheese [34] and Baijiu [35], is synthesized through the microbial metabolism of L-leucine in raw materials.

Isoamyl alcohol, as an advanced alcohol, is a crucial factor that forms the flavor quality of Baijiu and beer and can impart fruit and alcohol aroma to the product [36]. The right amount of isoamyl alcohol can make the taste of Baijiu more mellow. Its content was closely related to the type and content of amino acids and the glucose content in raw materials, mainly obtained by the conversion of leucine in raw materials through the amino acid anabolic/catabolic pathway or through the glycolysis of glucose and transamination reaction of valine [37].

Hexanal, a major product of fat oxidation, is formed during the oxidation of linoleic acid via 13-hydroperoxides and increases during the storage of fats and oils [38]. Previous studies have shown that lipoxygenase activity affects the amount of hexanal odor (grassy, fruity) during lipid oxidation; therefore, the decrease in hexanal content in PSP samples compared to SP may be due to the inactivation of lipoxygenase [39].

The key volatile flavor compounds specific to the SP sample were nonanal, heptanol, ethyl pentanoate, homotrimethylbenzene, 1-hepten-3-one, ethyl 2-methylbutyrate, and 1-octen-3-ol. Nonanal with a rose and citrus aroma is mainly produced by the raw materials’ oxidation of unsaturated fatty acids such as oleic, linoleic, linolenic, and arachidonic acids. Heptanol is primarily produced as a reaction product of lipid oxidation. Most of the esters in the samples had fruity or floral aromas, formed in part by the esterification of different alcohols and acids, with ethyl valerate and ethyl 2-methylbutyrate both having apple-like fruit aromas and being key volatile compounds in sausages and Baijiu [40]. 1-Hepten-3-one is considered to be the main reason for the mushroom flavor of foods, along with a geranium-like aroma. 1-Octen-3-ol is an aliphatic unsaturated alcohol that generally presents a mushroom aroma, has been detected in various fermented foods, and is also considered a key source of flavor in fermented foods such as soy sauce and soybean paste.

Key aroma compounds unique to the PSP samples included 4-heptanone, 2-pentylfuran, methanethiol hexanoate, isoamyl acetate, 3-methyl-4-heptanone, and isovaleraldehyde. Ketones, typically generated through the microbial fermentation-mediated degradation of lipids and amino acids, contribute to fermented foods’ fruity and hazelnut aromas. A study by Kiefl et al. [41] demonstrated that moderate levels of 3-methyl-4-heptanone contribute to hazelnuts’ desirable roasted and nutty flavor. 2-pentylfuran, whose flavor profile is butterscotch and floral, is considered to be the primary source of soy flavor in legumes and is mainly generated by enzymatic oxidation reactions caused by lipoxygenases, the content of which tends to decrease when fermented by microorganisms [23]. Although fermented food flavor compounds typically have low levels of sulfur compounds, their low flavor threshold allows them to influence the overall flavor profile significantly, even at low concentrations. They served as crucial flavor-contributing components in Baijiu. S-Methyl thiohexanoate imparted nutty and roasted meat aromas to the PSP samples, presumably from the formation of sulfur-containing amino acids in PSM released by an enzymatic reaction during fermentation and further reacted with reducing sugars by carboamidation. Isoamyl acetate is the main source of beer and brewing vinegar aroma, with a fruity flavor, generally formed by raw materials in the enzymatic action of alcohol acyltransferase. Isovaleric aldehyde is produced through the reaction between leucine in the raw material and α-dicarbonyl compounds, which are intermediate products of the Maillard reaction in the post-fermentation process. This reaction is associated with Strecker degradation and contributes to the chocolate and fat aromas. Compared to soybean, PSM exhibited a higher leucine content and underwent more Strecker degradation reactions, leading to an elevated isovaleraldehyde content in the PSP sample. Zhang et al. [42] showed that isovaleraldehyde is the main compound affecting fermented soy products’ flavor and taste.

## 4. Conclusions

In this study, peony seed meal was applied as one of the primary raw materials, along with soybean and flour, to produce fermented soybean paste. The optimal ratio of soybean to PSM was 7:3, at which neutral protease, amylase, and glucoamylase reached maximum enzyme activity at the first 48 h, with a value of 1177.85, 686.58, and 1564.36 U/g, respectively. It was observed that during post-fermentation, the addition of PSM to the soybean paste maintained the enzyme activities at a relatively high level overall. The soybean paste supplemented with PSM contained 80 species and 11 key flavor compounds, with 4-heptanone, 2-pentylfuran, methanethiol ester caproate, isoamyl acetate, 3-methyl-4-heptanone, and isovaleraldehyde as its exclusively key volatile compounds. More work is now in progress to further evaluate the influence of PSM addition on the bioactive and health-promoting properties of soybean paste. At the same time, a full sensory evaluation will be conducted. The study provided a new idea for enriching the flavor and nutritional qualities of fermented soybean products and demonstrated the potential application of PSM in food production.

## Figures and Tables

**Figure 1 foods-12-03184-f001:**
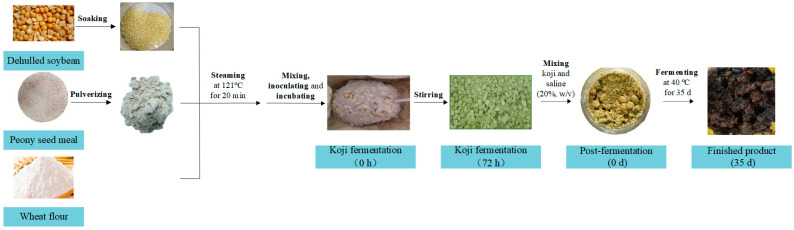
Flowchart of the fermentation process for the production of PSM soybean paste.

**Figure 2 foods-12-03184-f002:**
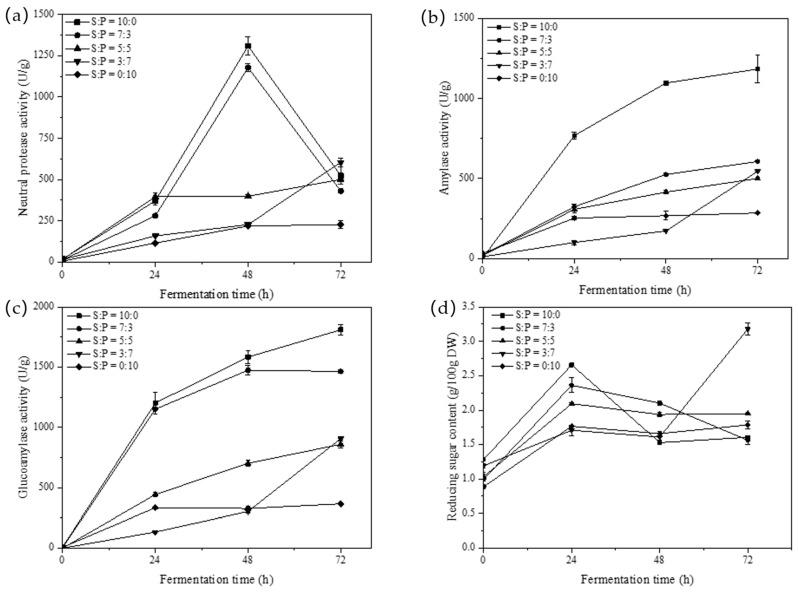
Changes in fermentation characteristics during the koji fermentation process. (**a**) Neutral protease activity; (**b**) amylase activity; (**c**) glucoamylase activity; and (**d**) reducing sugar content (S: Soybean; P: Peony seed meal).

**Figure 3 foods-12-03184-f003:**
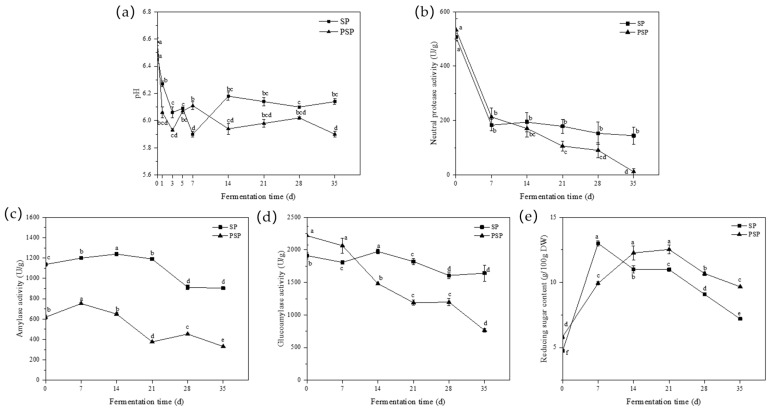
Changes in fermentation characteristics during the post-fermentation process. (**a**) pH; (**b**) neutral protease activity; (**c**) amylase activity; (**d**) glucoamylase activity; and (**e**) reducing sugar content (SP: Soybean paste; PSP: Peony seed meal paste). Different letters in each figure indicate statistically significant differences (*p* < 0.05).

**Figure 4 foods-12-03184-f004:**
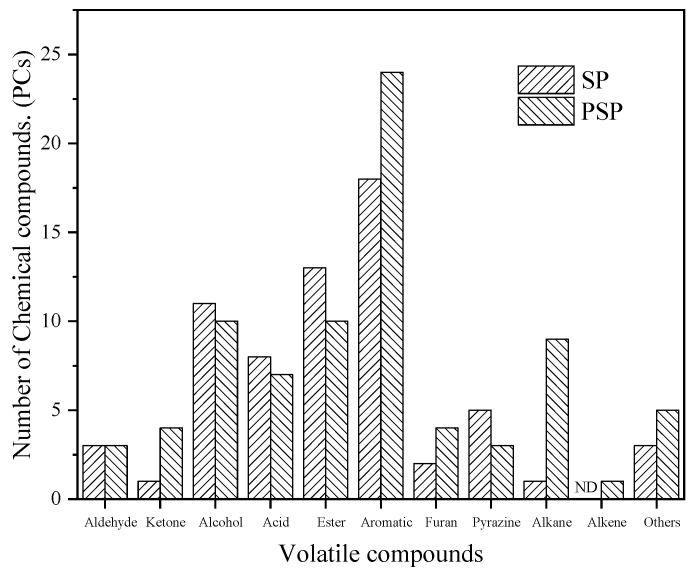
Differences in the types of volatile flavor substances of different soybean paste samples.

**Figure 5 foods-12-03184-f005:**
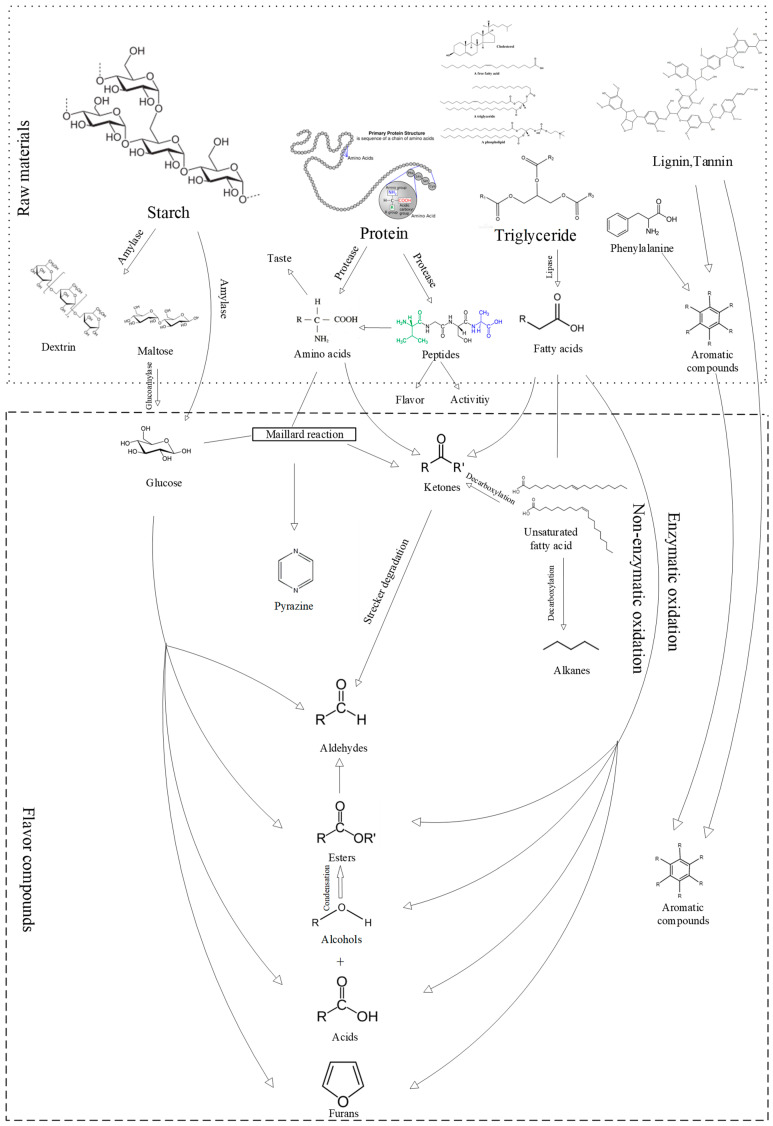
Related pathways involved in the formation and transformation of flavor compounds.

**Table 1 foods-12-03184-t001:** Flavor compounds of peony seed meal.

Flavor Compound	Odor Threshold (μg/kg) ^1^	Aroma Descriptors ^2^	Concentration (μg/kg)	OAVs
*N*-Methyl-2-methoxyethanamine	- ^3^	-	12.26 ± 2.97	-
Isovaleraldehyde	7	chocolate, peach	3.78 ± 0.24	0.17732
Hexanal	5	grass, fruity	26.13 ± 3.66	0.731468
*trans*-Cyclohexene carbonate	-	banana, cheesy	5.32 ± 2.47	-
(*E*)-2-hexanal	398.1	banana, cheesy	436.15 ± 34.99	0.339077
(*Z*)-2-pentenol	720	cherry, narcissus	14.72 ± 2.28	0.008726
2-Ethyl-4-methylpentanol	-	-	9.35 ± 1.19	-
(*E*,*E*)-2,4-Hexadienal	1.8	floral citrus	4.66 ± 0.78	2.102117
Acetic acid	180,000	sour vinegar	10.9 ± 1.02	1.12 × 10^−5^
Benzaldehyde	24	almond, cherry	128.79 ± 19.37	2.07227
Benzoic acid, methyl ester	7.5	cananga	34.89 ± 7.49	2.332165
2-Isopropoxy-ethylamine	-	-	4.48 ± 0.24	-
Benzofuran, 4,7-dimethyl-	-	-	183.26 ± 25.20	-
Benzyl alcohol	2546.21	floral rose	230.7 ± 20.22	0.02365
Benzoic acid	1000	faint balsam	587.58 ± 91.18	0.291178
18-Crown-6	-	-	15.15 ± 0.33	-
*n*-Hexadecanoic acid	10,000	waxy, fatty	75.01 ± 17.42	0.005742

^1^ Obtained by reviewing the literature; ^2^ flavor profile descriptions were obtained from http://www.thegoodscentscompany.com (accessed on 4 October 2022); https://www.chemicalbook.com (accessed on 4 October 2022), ^3^ Marked “-” indicated that not detected or relevant information was not retrieved, the same as below.

**Table 2 foods-12-03184-t002:** Flavor compounds with odor activity values (OAVs) ≥ 1 in PSP and SP samples.

Flavor Compound	Odor Threshold (μg/kg)	OAVs		Aroma Descriptors
		SP	PSP	
Ethyl isovalerate	0.1	24.02	82.44	apple, pineapple
Isovaleric acid	12	9.83	3.21	chess
Hexanal	5	6.76	1.82	leafy, fruity
Phenylacetaldehyde	6.3	4.90	30.90	floral hyacinth
3-Methyl-1-butanol	4	3.83	18.77	whiskey, fruity
1-Nonanal	1.1	4.73	-	rose, orange
*n*-heptanol	5.4	3.78	-	leafy, peony
Ethyl valerate	0.3	2.22	-	apple, pineapple
Mesitylene	3	2.00	-	-
1-Hepten-3-one	0.04	23.05	-	-
Ethyl 2-methylbutyrate	0.06	20.23	-	green apple
1-Octen-3-ol	1.5	12.06	-	mushroom
4-Heptanone	8.2	-	2.23	chess
2-Pentylfuran	5.8	-	8.34	butter, flower
S-Methylhexanethioate	0.3	-	22.33	cocoa, roasted meaty
Isoamyl acetate	0.15	-	125.99	fruity
3-Methyl-4-heptanone	0.05	-	157.76	fruity, hazel
Isovaleraldehyde	1.1	-	241.89	chocolate, fatty

## Data Availability

The data used to support the findings of this study can be made available by the corresponding author upon request.

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
