# Peer review of "Effect of Peony (Paeonia ostii) Seed Meal Supplement on Enzyme Activities and Flavor Compounds of Chinese Traditional Soybean Paste during Fermentation"

_foods, 2023, doi:10.3390/foods12173184_

Round 1

Reviewer 1 Report

The text is interesting, contains a scientific and practical aspect.

The introduction is sufficient and up-to-date, the experiment is logically planned and clearly described.

Corrections or explanations are necessary in several places:

Line 204 “In the koji fermentation stage, glucoamylase hydrolyzes polysaccharides such as maltose and dextrin into oligosaccharides” – This sentence needs correction, maltose can not be hydrolyzed into oligosaccharides

Figures 2 and 3: the symbols S, P, SP, PSP could be explained next to the graphs because the reader does not remember and has to return to the text

Line 251: “Additionally, a close correlation was observed between the decreasing trends of amylase and glycosylase activities within the same sample”: should it not be glucoamylase??? what kind of correlation, is it calculated somewhere?

Line 281: These products underwent a series of conversions within the branching acid metabolic pathway or phenylalanine secondary metabolism, ultimately leading to the production of aromatic secondary metabolites. – references needed

Figure 4 – amount on axis Y – needs units

Reviewer 2 Report

Researchers have investigated the impact of addition of by-product obtained from peony seeds after oil extraction on koji (soybean inoculated with Aspergillus oryzae- traditional Chinese food) enzyme activities (neutral protease, amylase, glucoamylase) and content of volatile compounds. Content of reducing sugars and odor activity values were also investigated. Literature studies using the Web of Science database have shown that no similar studies have been conducted in the past. The topic is original and relevant in the field. Methodology of investigations and references are correct. As the authors conclude “The study provided a new idea for expanding the sensory and nutritional qualities of fermented soybean products and demonstrated the potential application of PSM in food production”. I agree with it. This is the most important finding resulting from the research. It is likely that the drawings will require improvement as they will not fit on the page to be readable. Whether it is necessary to provide a CAS number. In compound names, "E", “N”, “n”, “tert” etc. should be written in italics. In Figure 5, the furan formula is drawn strangely, should be drawn like pyrazine. All chemical formulas should be drawn in the same graphics program. Line 480 is: “1-octen-3-ol”, it should be “1-Octen-3-ol”. Line 75: lack of PSM mass. Lines 79-80: lack of standardization of the inoculation process.

Reviewer 3 Report

The purpose of this study is to investigate the effect of PSM addition on the physicochemical and fermentation characteristics of soybean paste. Due to the addition of PSM, the quality of the fermented paste was improved, and the taste was enriched. Sensory evaluation should also be presented because the improvement of taste and aroma imparts a sensory function by the combination of analyzed compounds. There are the following problems.

1. Line 113. When citing for the first time, write the full DNS name.

2. Amylase activity and glucoamylase activity were measured by the DNS method, and maltose and glucose were used as standards, respectively. To measure Glucoamylase activity, glucose content should be measured with a glucose assay kit or HPLC. An accurate measurement method for glucoamylase activity analysis must be used.

3. In Koju production, the degree of proliferation of Asp.oryzae involved in fermentation is important. Therefore, cell mass analysis during the fermentation process should be performed. Among the methods for measuring cell mass, the method of analyzing glucosamine (Agric. Biol. Chem., 41 (4), 619 624, 1977) is commonly used.

3. Please enter the statistical analysis results in Fig. 2 and 3.

4. Fig. 4 explains the difference in the types of compounds detected in SP and PSP, so the amount on the Y axis does not seem to match. Change it to an appropriate expression, such as Number of Chemical compounds.

5. It was concluded that the flavor components of soybean paste were enriched and improved by using PSM. Flavor improvement is important in the sensory aspect by the combination of flavor components. Therefore, a sensory evaluation should also be presented to substantiate the author's assertion.

Round 2

Reviewer 3 Report

Although the author's opinion on the review was sufficiently presented in this paper, there are still some that have not been reflected.

1. As claimed by the authors, in many papers, gluco-amlylase and amylase activities are measured by the DNS method, and then glucose or maltose is measured as a standard substance. However, measuring these two activities requires a more detailed assay. At a minimum, gluco-amlyase needs to measure the enzymatic product.

2. The results in Fig 2 still do not reflect statistical treatment.

3. The change in cell mass is an essential metric for the fermentation process. Measurement results must be presented. Changes in cell mass and changes in enzyme activity are often different. Since aspergillus, not traditional koji, was used as a starter, cell mass analysis should be performed.

4. The authors considered that the characterization of flavor substances by HS-SPME combined with GC-MS may reflect that the addition of PSM can enrich flavor and sensory qualities. However, this is the result of flavor component analysis, and the abundance of flavor components and preference are different things. Being rich in flavor is not necessarily desirable. Therefore, I cannot agree with the conclusion that it will be beneficial to improve the quality and enrich the taste of traditional fermented paste and at the same time promote PSM as a valuable resource for fermented foods.
